# Neural and computational processes underlying dynamic changes in self-esteem

Geert-Jan Will[1,2]*, Robb B Rutledge[1,2], Michael Moutoussis[1,2], Raymond J Dolan[1,2]

[1]Wellcome Trust Centre for Neuroimaging, University College London, London, United Kingdom; [2]Max Planck UCL Centre for Computational Psychiatry and Ageing Research, University College London, London, United Kingdom

**Abstract** Self-esteem is shaped by the appraisals we receive from others. Here, we characterize neural and computational mechanisms underlying this form of social influence. We introduce a computational model that captures fluctuations in self-esteem engendered by prediction errors that quantify the difference between expected and received social feedback. Using functional MRI, we show these social prediction errors correlate with activity in ventral striatum/subgenual anterior cingulate cortex, while updates in self-esteem resulting from these errors co-varied with activity in ventromedial prefrontal cortex (vmPFC). We linked computational parameters to psychiatric symptoms using canonical correlation analysis to identify an 'interpersonal vulnerability' dimension. Vulnerability modulated the expression of prediction error responses in anterior insula and insula-vmPFC connectivity during self-esteem updates. Our findings indicate that updating of self-evaluative beliefs relies on learning mechanisms akin to those used in learning about others. Enhanced insula-vmPFC connectivity during updating of those beliefs may represent a marker for psychiatric vulnerability.

DOI: https://doi.org/10.7554/eLife.28098.001

*For correspondence: gjwill@gmail.com

**Competing interests:** The authors declare that no competing interests exist.

## Introduction

A positive sense of the self is the bedrock of mental health and well-being (*Orth et al., 2012*; *Trzesniewski et al., 2006*). Low self-esteem is a vulnerability factor for a range of psychiatric problems, including eating disorders (*Button et al., 1996*; *Vohs et al., 2001*), anxiety disorders (*Sowislo and Orth, 2013*) and depression (*Orth et al., 2008*; *Orth et al., 2009*). Classical theories in psychology view self-esteem as an internalization of actual and imagined appraisals from close others across development (*Cooley, 1902*; *Leary et al., 1995*; *Mead, 1934*). Indeed, an enduring sense of self-worth (referred to as 'trait self-esteem') reflects an accumulation of past appraisals from others (*Cole et al., 2001*; *Felson and Zielinski, 1989*; *Gruenenfelder-Steiger et al., 2016*; *Harter, 1983*; *Ladd and Troop-Gordon, 2003*), while momentary feelings of self-worth ('state self-esteem') are highly responsive to positive and negative social evaluations (*Denissen et al., 2008*; *Gerber and Wheeler, 2009*; *Thomaes et al., 2010*). Despite its importance for mental health, we lack a mechanistic understanding of how self-esteem depends on social evaluation. Here, using a novel social evaluation task, in combination with computational modeling and functional magnetic resonance imaging (fMRI), we characterize computational and neural processes underpinning changes in self-esteem.

A candidate neural substrate for integrating social evaluation with self-evaluation is ventromedial prefrontal cortex (vmPFC), given evidence that being evaluated by others (*Dalgleish et al., 2017*; *Gunther Moor et al., 2010*; *Somerville et al., 2010*), and evaluating the self (*Chavez et al., 2016*; *D'Argembeau et al., 2012*; *Hughes and Beer, 2013*; *Kelley et al., 2002*) activates subregions of

**eLife digest** Self-esteem – our evaluation of our own worth – is shaped by what other people think of us. It increases when others appreciate and value us, and decreases when we are rejected and start to question our own worth. Maintaining a positive sense of self is crucial for mental health and well-being. People with low self-esteem are more likely to develop psychiatric conditions, such as anxiety disorders, eating disorders and depression. Despite its importance for mental health, it was not known how the brain accumulates social feedback to determine our self-esteem.

To address this question, Will et al. developed a computational model that precisely predicts how self-esteem changes from moment to moment as people learn what others think of them. Activity in the brain was measured while young adults received approving or disapproving feedback from peers who had seemingly viewed their online character profile. After every second or third peer judgment, participants reported their current level of self-esteem.

Will et al. found that self-esteem depended both on whether other people liked the participants and on whether they were liked or disliked more than expected. Self-esteem decreased the most when participants received negative feedback from someone they expected to receive positive feedback from. The model then identified signals in specific parts of the brain that explain why self-esteem goes up and down according to the feedback received. Moment-to-moment changes in self-esteem correlated with activity in the ventromedial prefrontal cortex, which is a brain region important for valuation. Will et al. combined the model with responses to questionnaires that assessed psychiatric symptoms, and showed that vulnerable individuals had elevated responses in a part of the brain called the anterior insula. In vulnerable individuals, activity in this region of the brain was strongly coupled to activity in the part of the prefrontal cortex that explained changes in self-esteem.

A better understanding of the brain mechanisms that mediate a decline or improvement in self-esteem may help to find more effective treatments for a range of mental health problems.
DOI: https://doi.org/10.7554/eLife.28098.002

vmPFC. The vmPFC comprises multiple distinct cytoarchitectonic zones including parts of the anterior cingulate cortex (ACC; e.g., Brodmann areas 24 and 32) and orbitofrontal cortex (OFC; e.g., Brodmann areas 11, 13, 14) (*Haber and Knutson, 2010*). These subregions have distinct connectivity profiles (*Neubert et al., 2015*) and are thought to show specialization for cognitive functions important in valuation (*Bartra et al., 2013*; *FitzGerald et al., 2009*; *Lebreton et al., 2015*), social learning (*Apps et al., 2016*; *Hill et al., 2016*; *Suzuki et al., 2012*) and their intersection (*Apps and Ramnani, 2017*; *Garvert et al., 2015*; *Nicolle et al., 2012*). A recent study showed that the perigenual ACC (pgACC) tracks a history of one's own success and failures in a social context, while dorsomedial prefrontal area 9 tracked the performance history of an interaction partner (*Wittmann et al., 2016*). Crucially, activity in pgACC was higher in individuals whose subjective evaluation of their own performance was affected more strongly by their actual performance, making this region a candidate substrate for online updating of self-esteem.

Value representations in vmPFC are updated by teaching signals in the form of reward prediction errors that mediate an effect on vmPFC (*Frank and Claus, 2006*; *Garvert et al., 2015*; *Hampton et al., 2006*; *Jocham et al., 2011*; *Pasupathy and Miller, 2005*). By applying a computational modeling approach that has been shown to explain changes in subjective well-being during value-based decision-making (*Rutledge et al., 2014*; *2015*), we test whether self-esteem is dynamically updated during social evaluations through the cumulative impact of 'social approval prediction errors' (i.e., the difference between expected and received social feedback). More specifically, using fMRI, we could test whether dynamic self-esteem updates are reflected in vmPFC activity, and whether individual variation in neural encoding of social approval prediction errors (SPEs) explains inter-individual differences in self-esteem. Through combining computational parameters with psychiatric symptoms in a single multivariate analysis, we identified a dimension of 'interpersonal vulnerability'. Vulnerability, so defined, was associated with enhanced SPEs in anterior insula and greater insula-vmPFC functional connectivity during self-esteem updates, suggesting potential biomarkers for psychiatric vulnerability.

## Results

### Computational processes underlying dynamic changes in self-esteem

We scanned 40 participants (mean age = 23.4, *SD* = 3.3, 14 male) while they performed a newly designed social evaluation task. In this task participants received approval and disapproval feedback. This feedback was ostensibly derived from 184 strangers who had viewed an online character profile each participant had uploaded to an online database, one week prior to the experiment (see *Supplementary file 1* for details). Participants' expectations about feedback were manipulated by sorting raters into four groups based on their ostensible overall approval rates toward all participants in the experiment. All participants received approval feedback in 85%, 70%, 30%, and 15% of the trials, spanning the first to fourth quartile rater groups respectively (see *Figure 1*). Participants were not informed about these exact probabilities, but to better orientate them they learned the rank ordering of the four groups prior to performing the task.

On each trial, subjects were provided with the name of a rater and a color cue that indicated the rater's group (see *Figure 1*). They were then asked to predict whether the rater liked or disliked them. After a delay, they received approval (in the form of a thumbs up symbol), or disapproval (in the form of a thumbs down symbol) feedback. After every 2–3 choice trials, participants reported their current level of self-esteem by answering how good they felt about themselves at that exact moment using a visual analogue scale (see Materials and methods for further details on the task).

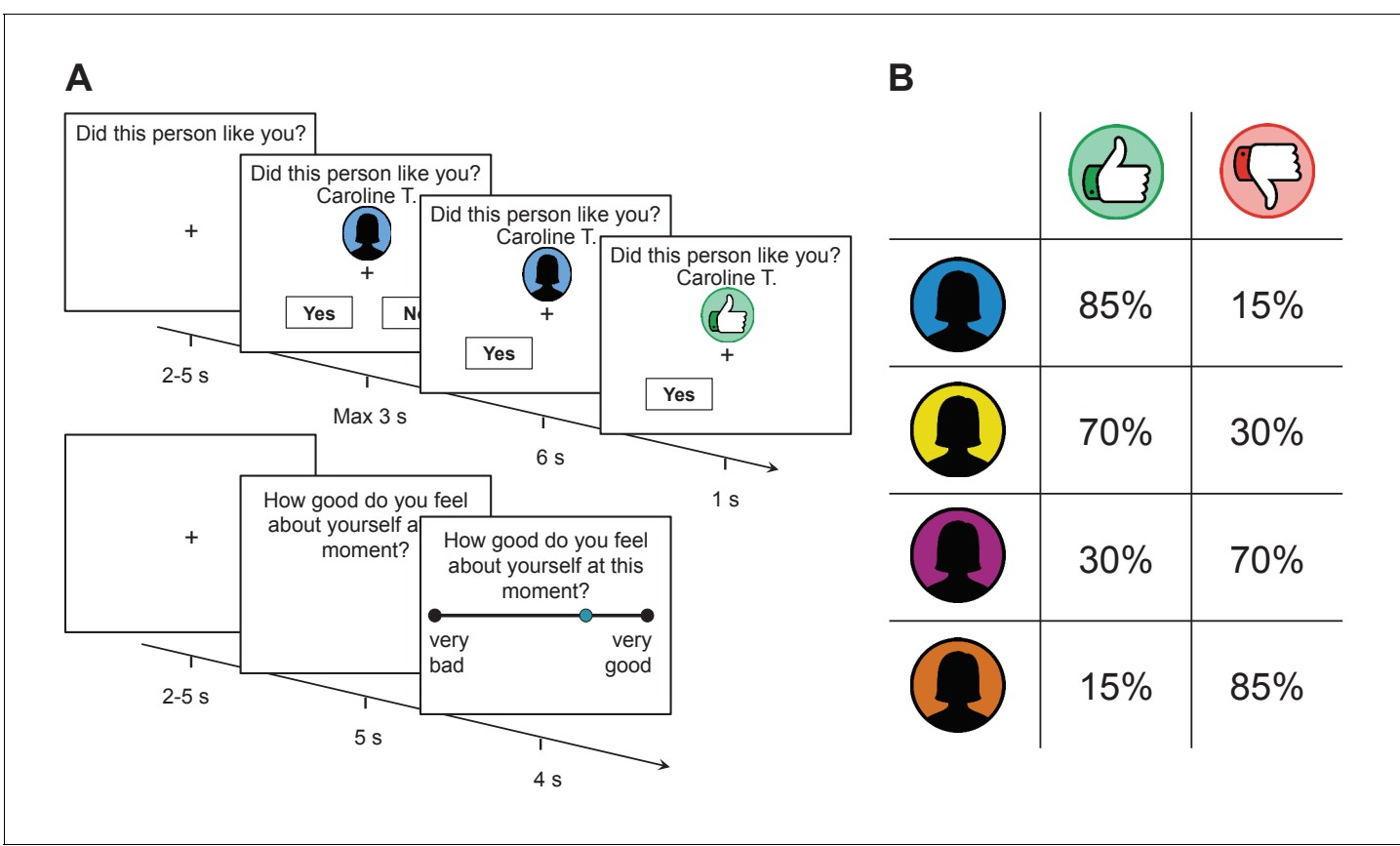

**Figure 1.** Task structure and feedback probabilities. (A) Participants were provided with a visual cue that indicated which group a rater belonged to (assigned according to their overall disposition to provide approving or disapproving feedback; see panel on the right). They then made a prediction as to whether the rater would like or dislike them before receiving feedback. After every 2–3 trials, participants were asked to indicate their current level of self-esteem. (B) Probability of receiving approval or disapproval feedback was dependent on the rater's group (signaled by a color cue). Participants received approving feedback in 85%, 70%, 30%, and 15% of the trials. Group colors were randomized across participants.
DOI: https://doi.org/10.7554/eLife.28098.003

First, we established that participants adapted their predictions about being liked based on a rater's group membership. A repeated-measures ANOVA with group (4 levels: 85%, 70%, 30% and 15% profiles liked) as a within-subjects factor and percentage predictions of being liked as a dependent variable showed a main effect of group, $F(3,117) = 209.47$, p<0.001, $\eta_p^2$=0.843. Pairwise comparisons showed that participants predicted they would be liked more by raters from group 1 (95%) than group 2 (87%, p<0.001), more by raters from group 2 than 3 (30%, p<0.001) and more by raters from group 3 than 4 (13%, p<0.001; see *Figure 2A*). These choice patterns were reliably predicted by a Rescorla-Wagner reinforcement-learning model (*Rescorla and Wagner, 1972*) fitted to participants' choice behavior (mean $r^2 = 0.40 \pm 0.27$; mean ± SD; see Materials and methods for details on this model). The modeling results were consistent with participants using SPEs (the difference between received feedback and expected social feedback), weighted by a learning rate, to update their expectations about approval from raters from each of the four groups (ESV), which in turn guided their predictions about being liked (see *Figure 2*). The model includes a bias parameter ($ESV_0$) that captures persistent beliefs about the probability of being liked or disliked. Participants with a larger $ESV_0$ persisted in predicting they would be liked by raters from groups for whom they had a negative ESV (e.g., raters from groups 3 or 4).

Next, we tested the hypothesis that dynamic changes in self-esteem depend on both the valence of social feedback and expectations about feedback. If SPEs explain changes in self-esteem better than outcome valence alone, trial-by-trial changes in self-esteem should correlate positively with outcome valence and negatively with expectations (*Behrens et al., 2008*). We found that, after regressing out the positive effect of outcome valence (r = 0.18, p<1×10$^{-9}$), there remained a significant negative correlation with expectations (r = −0.06; p=0.012) (see *Figure 2B*). We formally modeled the cumulative impact of SPEs on moment-to-moment variation in self-esteem using exponential kernel regression models. Parameters were fit to both choice behavior and self-esteem ratings in individual participants. Our winning model (*Equation 1*) successfully captured dynamic changes in self-esteem at the level of the individual ($r^2 = 0.32 \pm 0.24$; mean ± SD; see *Figure 2*). We chose this model as it outperformed a range of alternative plausible models, including a model that accounted for the valence of social feedback, but did not feature expectations about approval ('Outcome valence only' model 6; see *Table 1*).

$$\text{Self-esteem}(t) = w_0 + w_1 \sum_{j=1}^{t} \gamma^{t-j} SPE_j + \epsilon \qquad (1)$$

For each trial ($t$), we entered a term for baseline self-esteem throughout the task ($w_0$) and a term capturing the weight of SPEs ($w_1$) into the equation. Expectations about social approval were estimated using the previously mentioned reinforcement-learning model. Through the inclusion of a forgetting factor ($\gamma$) the influence of SPEs was allowed to decay exponentially in time, such that recent events had greater impact than earlier events. The Gaussian noise term ε ~N(0, σ) allowed Equation 1 to serve as a generative model of self-esteem.

The average learning rate $\eta$ (involved in updating expectations about approval from the raters) was 0.04 ± 0.07 (mean ± SD) and the average forgetting factor $\gamma$ (involved in updating self-esteem) was 0.65 ± 0.35 (mean ± SD; see *Table 2* for means and standard deviations of all model parameters). This indicates that SPEs induce rapid changes in feelings about the self, but impact learning about the probability of approval from the four groups relatively slowly.

Bayesian model comparison showed our favored model outperformed alternative models (see *Table 1*), including: (1) models without a response bias parameter ($ESV_0$) which captured persistent beliefs about the probability of being approved or disapproved, (2) models without learning (including a model where participants had correct initial beliefs about approval expectations for the four groups that were not dynamically updated based on feedback), (3) a model with a separate expectations term to test whether expectations have additional effect on self-esteem, above and beyond their effect captured by the SPE term and (4) a model that only took the valence of feedback into account, but did not feature expectations about approval (see methods for details on models).

This comparison provides support for an hypothesis that self-esteem is sensitive to a cumulative impact of recent prediction errors arising out of expectations concerning social approval. Furthermore, these expectations are not stable, but are dynamically updated and depend on persistent beliefs about approval. Finally, changes in self-esteem are better described by a model without a

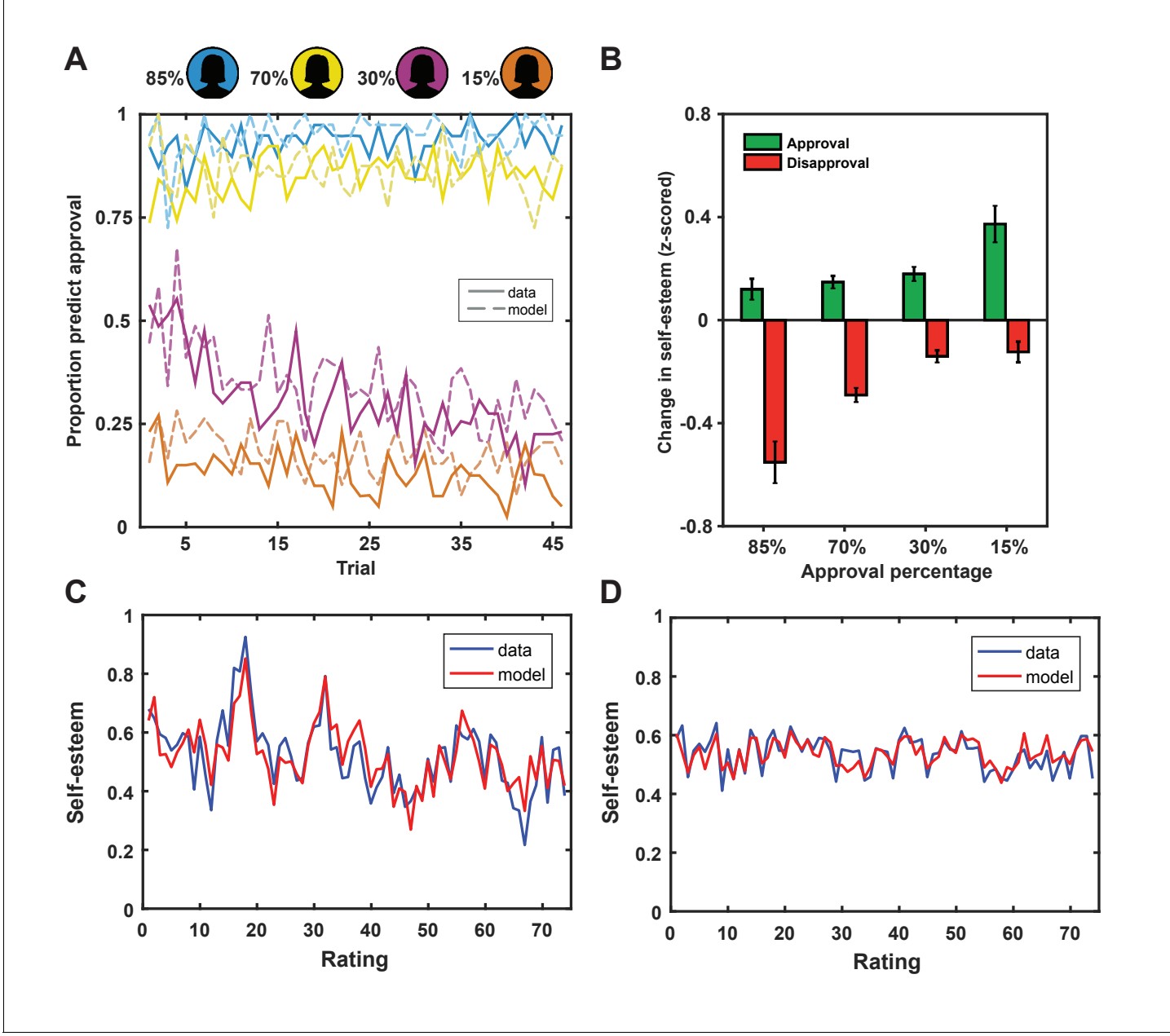

**Figure 2.** Behavioral results. (**A**) Participants adapted their predictions about being liked by a rater based on the rater's group membership (solid lines; colors indicate approval probability of group) and our learning model explained subject predictions (dotted lines). (**B**) Changes in self-esteem in response to feedback depended on social approval expectations that differed for the different groups, such that self-esteem depended on both valence (approval vs. disapproval) and the probability of approval. The largest self-esteem increases occurred with approval feedback from the 15% group, the most surprising positive feedback. The largest self-esteem decreases occurred with disapproval feedback from the 85% group, the most surprising negative feedback. Data are represented as mean ± standard error of the mean (SEM). (**C** and **D**) Self-esteem ratings over the course of the experiment in two exemplar participants (in blue) and predictions of our computational model (Equation 1; in red).

DOI: https://doi.org/10.7554/eLife.28098.004

The following figure supplements are available for figure 2:

**Figure supplement 1.** Results 'Other evaluation' Control Task.

DOI: https://doi.org/10.7554/eLife.28098.005

**Figure supplement 2.** Results Dictator Game.

DOI: https://doi.org/10.7554/eLife.28098.006

**Table 1.** Comparisons of fits of self-esteem models

Bayesian Information Criterion (BIC) measures are summed across all participants. Lower BIC values indicate a more parsimonious model fit. Mean squared error over self-esteem ratings indicates goodness of fit. k is the number of fitted parameters. Models 1–4 were fit to both choice behavior and self-esteem ratings and BIC measures comprise both the summed log likelihood of the model prediction over choice behavior and the summed log density of the model prediction over the self-esteem ratings. Models 5–6 were fit solely to the self-esteem ratings in order to allow for a fair comparison with a model without expectations (model 6), which by definition would not provide a good fit for the behavioral choice data. See Materials and Methods for details on the computational models.

| Model | k | Mean $r^2$ | Median $r^2$ | BIC | BIC-BIC$_{model1}$ |
|---|---|---|---|---|---|
| 1: Learning and positive bias | 9 | 0.31 | 0.27 | −633 | 0 |
| 2: Learning, but no bias | 8 | 0.29 | 0.25 | −378 | 255 |
| 3: Correct initial beliefs about approval | 7 | 0.25 | 0.22 | 409 | 1042 |
| 4: Separate term for expectations | 10 | 0.34 | 0.32 | −502 | 131 |
| Model | k | Mean $r^2$ | Median $r^2$ | BIC | BIC-BIC$_{model1}$ |
| 5: Free initial beliefs about approval | 5 | 0.32 | 0.31 | −5671 | 0 |
| 6: Outcome valence only | 7 | 0.23 | 0.18 | −5581 | 90 |

DOI: https://doi.org/10.7554/eLife.28098.007

separate effect of expectation, indicating that the effect of expectation on self-esteem – unlike in the case of mood (*Rutledge et al., 2014*; *2015*) - only operates through prediction errors realized at the moment that feedback is delivered. Results from a control experiment demonstrated that the

**Table 2.** Means and standard deviations for computational self-esteem parameters and psychiatric symptom measures.

| Computational self-esteem parameters | Mean (SD) |
|---|---|
| Baseline self-esteem ($w_0$) | 0.73 (0.16) |
| Average initial approval beliefs ($\frac{ESV_1^{(1)}+ESV_4^{(1)}}{2}$) | 0.64 (0.24) |
| Decision temperature ($T$) | 0.12 (0.35) |
| Sigma in gaussian noise term | 0.08 (0.04) |
| Weight on SPEs ($w_1$) | 0.04 (0.03) |
| Range initial approval beliefs ($ESV_1^{(1)} - ESV_4^{(1)}$) | 0.34 (0.29) |
| Bias parameter ($ESV_0$) | 0.42 (0.25) |
| Learning rate ($\eta$) | 0.04 (0.08) |
| Forgetting factor ($\gamma$) | 0.66 (0.35) |
| **Symptom questionnaires** | **Mean (SD)** |
| Trait Self-Esteem (*Rosenberg, 1965*) | 21.60 (5.00) |
| State Self-Esteem (*Heatherton and Polivy, 1991*) | 77.93 (10.23) |
| Self-Perception (*Neemann and Harter, 1986*) | 2.98 (0.61) |
| Narcissism (*Raskin and Terry, 1988*) | 11.26 (5.8) |
| State Anxiety (*Spielberger et al., 1970*) | 1.62 (0.47) |
| Trait Anxiety (*Spielberger et al., 1970*) | 1.93 (0.47) |
| Social Anxiety (*Liebowitz, 1987*) | 1.05 (0.88) |
| Rejection Sensitivity (*Downey and Feldman, 1996*) | 9.24 (2.96) |
| Fear of Negative Evaluation (*Carleton et al., 2011*; *Leary, 1983*) | 2.71 (0.94) |
| Depression (*Beck et al., 1996*) | 5.40 (5.47) |
| Depressed Mood (*Angold et al., 1995*) | 9.15 (7.24) |

DOI: https://doi.org/10.7554/eLife.28098.008

observed self-esteem changes in the fMRI task are specific to situations where the self is the object of evaluation and are unlikely to be the result of demand characteristics (see *Figure 2—figure supplement 1*).

## Linking computational self-esteem parameters to symptoms

To examine relationships between computational self-esteem parameters and symptoms linked to low self-esteem we performed a canonical correlation analysis (CCA; *Hair et al., 1998*). CCA finds the maximal correlation between a linear combination of one set of variables (in our case self-esteem parameters from our computational model) and a linear combination of another set (in our case symptoms linked to low self-esteem and interpersonal sensitivity measured using questionnaires; see Materials and methods for details). The CCA yielded one significant canonical dimension (Wilks's $\lambda = 0.01$, $F(99, 152.5) = 1.40$, p=0.029), which had a canonical correlation of 0.87 between computational parameters and symptoms. We labeled the dimension as 'interpersonal vulnerability' based on the constellation of positive and negative associations of the different computational parameters and symptom measures with the identified dimension (see *Table 2* and *Figure 3*).

With respect to symptoms, trait and state self-esteem showed a strong negative association with 'interpersonal vulnerability'. Symptoms of depression, social anxiety, and trait and state anxiety showed a positive association with 'interpersonal vulnerability'. As for the computational parameters, baseline self-esteem ($w_0$) and average initial approval beliefs showed a negative association with 'interpersonal vulnerability'. Weight on SPEs ($w_1$) and the range of initial approval beliefs showed a positive association with 'interpersonal vulnerability'. The results highlight that people with lower self-esteem and greater anxiety and depression symptoms have lower expectations about approval and greater self-esteem fluctuations in response to SPEs.

## Neural processes underlying dynamic changes in self-esteem

We first examined encoding of SPEs in a whole-brain regression analysis with trial-by-trial SPEs (inferred using our computational model and time-locked to feedback onset) as a parametric modulator. This analysis revealed SPEs correlated with activity in a cluster in bilateral ventral striatum extending into subgenual anterior cingulate cortex (sgACC; BA 25) (see *Figure 4A*; left peak coordinates −8, 21,–5; $t(39) = 4.50$; right peak coordinates 5, 20,–8, $t(39) = 5.42$; Z = 4.65, k = 1172, p=0.005, Family-wise Error [FWE] cluster-corrected). Next, we tested whether neural activity at feedback reflected self-esteem updates contingent on feedback. We regressed activity at feedback presentation against trial-by-trial updates in self-esteem (i.e., inferred using our computational model). This analysis revealed a significant cluster in ventromedial prefrontal cortex (vmPFC) with a peak in left medial OFC (Brodmann Area [BA] 14m) extending into pgACC (BA 32pl) (see *Figure 4B*; peak coordinates −6, 33,–15, $t(39) = 3.83$; Z = 3.51, k = 868, p=0.047, FWE cluster-corrected).

Our next analyses focused on testing whether individual differences in 'interpersonal vulnerability' are reflected in neural representations of SPEs and self-esteem updates. We used our CCA results to obtain subject-specific scores on the 'interpersonal vulnerability' dimension. These scores are the result of a new weighting of symptoms that maximally correlated with computational self-esteem parameters. First, we ran a whole-brain regression analysis with trial-by-trial SPEs as parametric modulator and 'interpersonal vulnerability' scores as a between-subject regressor to test which brain regions responded more strongly to SPEs in individuals who are more vulnerable relative to those who are less vulnerable. This revealed a positive association between 'interpersonal vulnerability' and activity in a left anterior insula cluster extending into inferior frontal gyrus (see *Figure 4C*; peak coordinates −44, 11, 9; $t(38) = 4.70$; Z = 4.15, k = 5463, p<0.001, FWE cluster-corrected).

Our behavioral finding of a positive association between vulnerability and self-esteem updating in response to SPEs, motivated us to test an hypothesis that vulnerability-related variation in self-esteem updating is mediated through variation in functional connectivity between the vmPFC and the insula region where SPE-related activity was modulated by vulnerability. Consequently, we examined functional connectivity between the insula (using a 6 mm sphere around the peak of the cluster from the previous analysis as a seed region) and the rest of the brain during self-esteem updates using a psychophysiological interaction (PPI) analysis (*Friston et al., 1997*; *O'Reilly et al., 2012*). This connectivity analysis showed that 'interpersonal vulnerability' correlated positively with functional coupling between the insula and a cluster in vmPFC with a peak in right medial OFC (BA 14m)

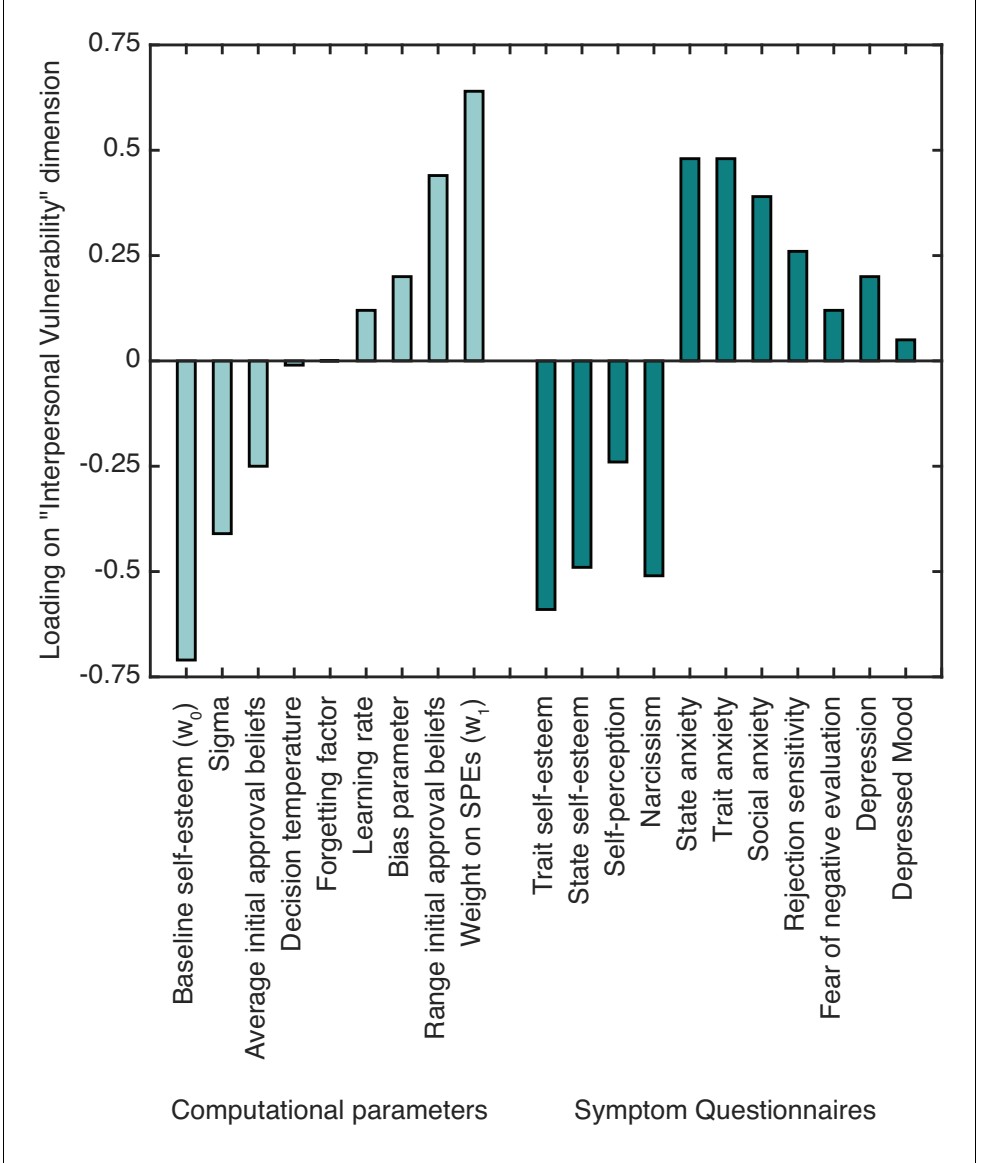

**Figure 3.** The standardized canonical coefficients for the 'interpersonal vulnerability' dimension across computational self-esteem parameters and psychiatric symptom measures.
DOI: https://doi.org/10.7554/eLife.28098.009

during self-esteem updates (see *Figure 4D*; peak coordinates 11, 32,–11; $t(38)$ = 6.27; Z = 5.16, k = 78570, p<0.001, FWE cluster-corrected). Interpersonal vulnerability did not correlate with SPE-related activity in the striatum/sgACC cluster (Spearman's ρ = 0.238, p=0.140) or updating-related activity in the vmPFC cluster (Spearman's ρ = −0.005, p=0.978). Thus, greater interpersonal vulnerability (i.e., more symptoms and amplified self-esteem parameters) is associated with both increased SPE responses in anterior insula and greater functional connectivity between the insula and the vmPFC during self-esteem updates. Together these results hint at potential mechanisms for vulnerability to psychiatric illness.

## Discussion

Self-esteem is shaped by what other people think of you (*Cooley, 1902*; *Denissen et al., 2008*; *Gerber and Wheeler, 2009*; *Leary et al., 1995*; *Mead, 1934*; *Thomaes et al., 2010*). Our findings

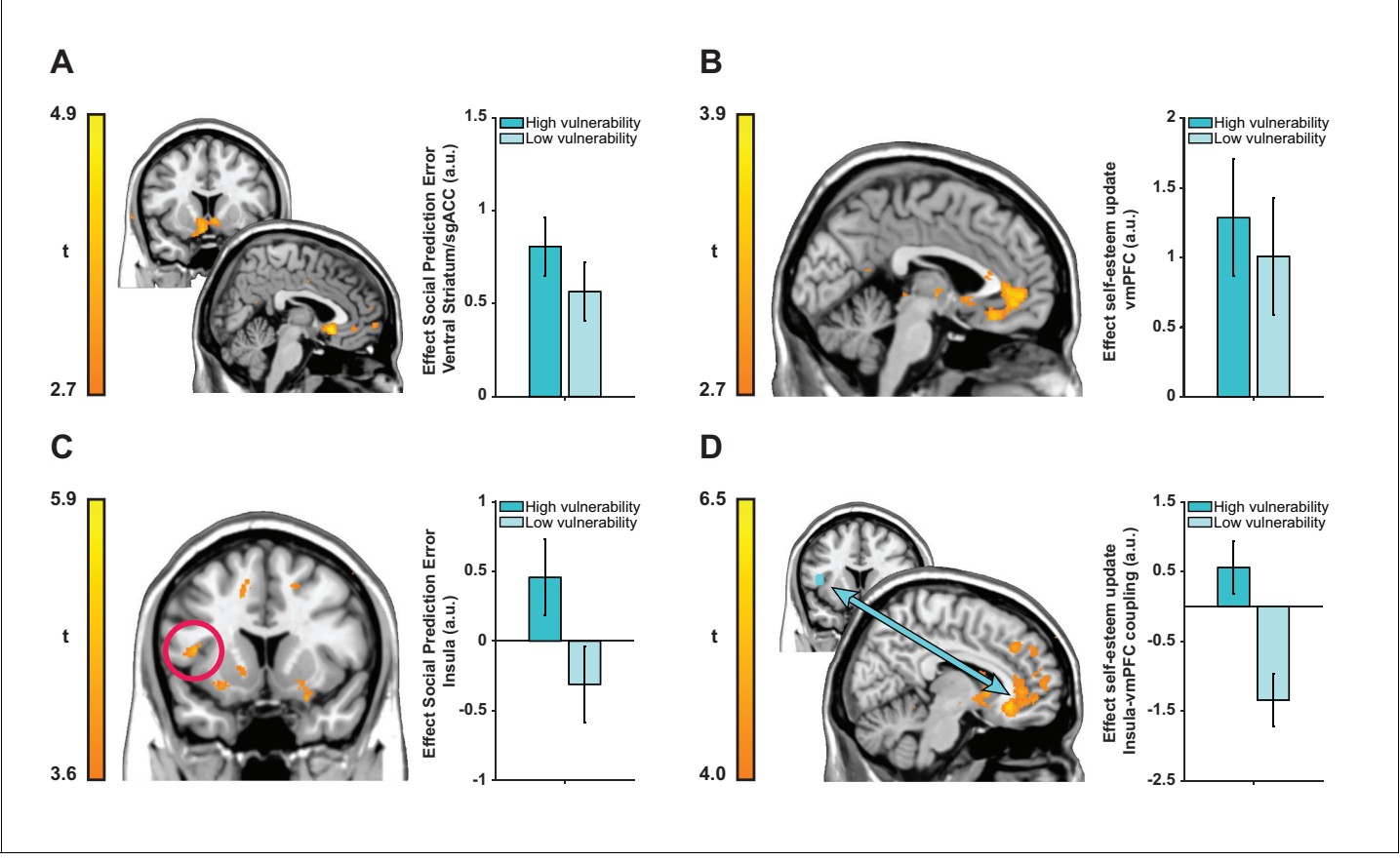

**Figure 4.** Neuroimaging results plotted separately for high and low 'interpersonal vulnerability' participants (based on median split) to facilitate interpretation. (**A**) Social prediction errors correlated with activity in a cluster in bilateral ventral striatum extending into sgACC. (**B**) Trial-by-trial updates in self-esteem upon receipt of feedback correlated with activity in vmPFC (BA 14m and BA 32pl). (**C and D**) Vulnerability modulated the expression of prediction error responses in left anterior insula (extending into inferior frontal gyrus) and insula-vmPFC coupling during self-esteem updates. Images are thresholded at t > 2.7 (panels **A** and **B**), t > 3.6 (panel **C**) and t > 4.0 (panel **D**) with no cluster-extent threshold for display purposes. Data are represented as mean ± SEM. See tables in ***Supplementary file 2*** for a full list of activations.
DOI: https://doi.org/10.7554/eLife.28098.010

reveal how this form of social influence on self-esteem is implemented in the brain. Our computational modeling results are consistent with people using prediction errors to learn what to expect from others and to update their self-esteem based on the outcome of these expectations. Using fMRI, we show that SPEs correlate with activity in ventral striatum and the sgACC, while self-esteem belief updates are reflected in vmPFC activity. The findings highlight that learning from social evaluative feedback and updating self-evaluative beliefs rely on learning mechanisms as seen in social and non-social reward learning at both an algorithmic (i.e., prediction error driven) and neural level (i.e., shared neural substrates in the striatum, ACC, and vmPFC).

Self-esteem has characteristics akin to a 'gauge of social acceptance' as articulated within 'Sociometer theory' (***Leary et al., 1995***). Our study confirms predictions made by sociometer theory and places this notion within a quantitative and neurobiologically grounded framework. A notable feature of our data is that it indicates that self-esteem is not a sociometer that merely maintains an ongoing tally of social acceptance, but is more akin to a read-out of the extent to which our social standing has undergone change recently. Our data indicate that forming accurate expectations about rejection exerts a buffering effect against expected rejection, fitting observations that self-esteem is more volatile in individuals with aberrant expectations about rejection (***Dandeneau and Baldwin, 2004***; ***Leary et al., 1995***). These findings lead to a new testable hypothesis, namely that self-esteem encompasses a form of learning signal that we use to gauge our social standing in new environments.

Our winning computational model entailed a mean forgetting rate such that state self-esteem depended on the most recent six appraisals acquired from raters. In contrast, people's beliefs about the global social milieu (related to estimates about how approving people are in the current social environment) accumulated slowly and were much more resistant to change. Thus, it appears that people use SPEs as an estimate of the *local* gradient of social approval that informs their experienced self-esteem, as well as for slowly updating beliefs about expectations of approval from the *global* social milieu. In this way, changes in self-esteem can be thought of as learning about the self, which is distinct but related to learning about others.

Learning about social approval not only has algorithmic similarity to learning about non-social stimuli (*Montague et al., 1996*; *Schultz, 2013*; *Sutton and Barto, 1998*), but also depends on similar neural circuitry. SPEs correlated with activity in a cluster including the ventral striatum and the sgACC. The striatum encodes prediction errors in learning about primary (*D'Ardenne et al., 2008*; *Hart et al., 2014*) and secondary rewards (*Caplin et al., 2010*; *Pessiglione et al., 2006*), including learning about social acceptance (*Jones et al., 2011*; *Jones et al., 2014*). While the ventral striatum has been shown to process both prediction errors about rewards for the self and other people, there is also evidence that sgACC exclusively encodes prediction errors about rewards for other people (*Lockwood et al., 2016*). Consistent with this notion, a recent social learning study showed that ventral striatum processes prediction errors about the accuracy of another person's advice, while a region in the sgACC/septum encodes prediction errors about the person's general level of trustworthiness (*Diaconescu et al., 2017*). The social approval prediction errors in our task drive changes in self-esteem in response to social rewards for self and at the same time guide learning about other people's general level of 'niceness'. The presence of a cluster spanning both striatum and sgACC in response to this multiplexed prediction error signal dovetails with these prior findings. A goal for future research is to disentangle the different contributions of ventral striatum and sgACC during learning about the self through interactions with others.

Based on prior work showing the influence of secondary reward (i.e., monetary) prediction errors on mood (*Rutledge et al., 2014*; *2015*), it is likely that our experimentally induced SPEs also affect mood. Like mood, self-esteem depends on expectations that lead to prediction errors. However, model comparison demonstrated a key difference between an established model of momentary variation in mood and the self-esteem model we introduce here. Mood increases at the moment in time when subjects know they may gain future rewards, even in the absence of feedback (*Rutledge et al., 2014*; *2015*). Our results show that self-esteem does not increase when individuals are in an environment with socially accepting others and, unlike for mood, model comparison does not support self-esteem models with a separate expectation term.

Given a wealth of studies on neurocomputational mechanisms supporting valuation (*Bartra et al., 2013*; *Rangel et al., 2008*), it is surprising how little is known about the computations supporting the most fundamental type of valuation, namely the evaluation of our own worth. Our results show that updates of self-worth are represented in a cluster in vmPFC with a peak in medial OFC (BA 14m), extending into pgACC (BA 32pl). Activity in the pgACC has been shown to increase and decrease in response to recent success and failure of the self, but not of others, in joint decision-making tasks (*Wittmann et al., 2016*). Our results extend the role of the pgACC in self-related processing by showing that this region not only keeps track of how well one is performing in a given task, but that it continuously updates a more general value ascribed to the self when learning how others value us. Neurons in adjacent subregion BA 14m show correlated tuning for reward size and reward probability in monkeys, suggesting that BA 14m neurons encode an integrated value signal (*Strait et al., 2014*). This is consistent with a large-scale meta-analysis of 81 human fMRI studies showing that activity in BA 14m correlates with subjective value ascribed to a range of primary and secondary rewards, both upon receipt of a reward and during choice formation (*Clithero and Rangel, 2014*). Adjacent subregions in vmPFC may thus integrate separate strands of information about current value assigned to the self in order to estimate self-value in the future.

Our results suggest that social approval may act on self-value representations in a manner similar to the effects of primary and secondary rewards on value representations about external stimuli. As in the latter, updates in self-esteem upon receipt of feedback co-varied with activity in vmPFC akin to updates of value attributed to external stimuli (*Behrens et al., 2008*; *FitzGerald et al., 2009*; *Rushworth et al., 2011*), strengthening the idea that self-evaluation may be reducible to valuation, but where now the object is the self (*D'Argembeau, 2013*). In this light the social modulation of a

self-esteem representation in vmPFC is also consistent with findings showing that the vmPFC integrates social with personal preferences to compute a new value of an object based on to the opinions of other people (*Campbell-Meiklejohn et al., 2017*).

By combining computational self-esteem parameters with measures of psychiatric symptomatology in a single multivariate analysis, we identified a dimension of 'interpersonal vulnerability'. Vulnerability was associated with low self-esteem, internalizing symptoms and self-esteem instability in response to SPEs. This was mirrored at a neural level by augmented SPE processing in anterior insula and a greater positive functional connectivity between the insula and a cluster in vmPFC with a peak in BA 14m. This suggests self-esteem instability may result from a greater malleability of self-value representations in vmPFC driven by prediction error signals arising from the insula. The location of the cluster in the insula shows striking overlap with findings from social anxiety patients during reappraisal of negative self-beliefs (*Goldin et al., 2009*) as well as findings in a range of anxiety disorders (*Etkin and Wager, 2007*). As such, increased responsivity to SPEs and greater insula-vmPFC coupling during self-esteem updates may represent neurobiological markers of a dimension of 'interpersonal vulnerability' that confers increased risk for a number of common mental health problems.

A question for future research is whether psychiatric patients, especially those suffering from internalizing disorders like depression or anxiety, fall at the extreme end of the 'interpersonal vulnerability' dimension that we identified. Such thinking is at the core of the Research Domain Criteria (rDOC; *Insel et al., 2010*) which aims at re-conceptualizing psychiatric nosology by identifying dimensions of biologically plausible trans-diagnostic markers. Our approach allows identification of a new weighting of questionnaire measures with sensitivity to individual differences in neural processes relevant to rapid changes in self-esteem. This questionnaire weighting, which relates to both our new self-esteem computational model and the neural responses to social feedback processing, might index risk of future mental health outcomes.

We demonstrate that state self-esteem can be conceptualized as a self-value representation in vmPFC, a representation that is dynamically updated through prediction errors resulting from violations of expectations about evaluative feedback. Inter-individual variation on a symptom dimension that cuts across traditional diagnostic categories mapped closely to indiviual differences in insula responses to social feedback and insula-vmPFC coupling during self-esteem updates. Our framework thus reveals fundamental mechanisms that underlie how we use social information when evaluating ourselves and holds promise as a trans-diagnostic predictor of psychiatric outcomes.

## Materials and methods

### Participants

We recruited forty-four participants through participant pools at University College London (UCL). Sample size was based on prior fMRI studies examining inter-individual differences in social feedback processing (*Powers et al., 2013*; *Somerville et al., 2010*). Exclusion criteria included a prior history of head injury, neurological or psychiatric disorder, color blindness, or being left-handed. We monitored participants using an eye tracker and we excluded participants who were shown to have fallen asleep during scanning (n = 4). The target sample comprised 40 participants (mean age = 23.3, *SD* = 3.2, 14 male) who were paid a fee of £8 per hour plus earnings based on an additional experiment after the MRI scans (Dictator Game; see *Figure 2—figure supplement 2*). Informed consent was obtained from every participant and experimental procedures were approved by the local research ethics committee.

### Procedure

Participants were invited to come to the lab seven days prior to the MRI experiment to create and upload a personal profile (see *Supplementary file 1*). As part of the cover story for the experiment, we showed them an online database and explained that we needed several days to receive a sufficient number of evaluations, and as a consequence the scanning session would take place seven days later. During this first session, participants also filled out a battery of questionnaires (see below). The MRI session included a training part during which participants learned, and were tested on, the structure of the fMRI task before going into the scanner. Subsequent to scanning they were given a set of additional experimental tasks (see *Figure 2 – figure supplement 1* and *2*).

## Social evaluation task

Participants performed a new social evaluation task, which was inspired by existing paradigms (*Eisenberger et al., 2011*; *Gunther Moor et al., 2010*; *Somerville et al., 2006*). In this task they received feedback indicating ostensible approval or disapproval from 184 strangers ('raters'; 92 males and 92 females). Participants were told that raters were sorted into four groups based on their overall approval rates toward all participants in the experiment. On each trial, participants were presented with the name of a rater and a color cue that indicated the rater's group, assigned according to overall approval rates (see *Figure 1*). After a jittered fixation display (uniformly distributed between 2 and 5 s), participants had 3 s to predict whether the rater approved or disapproved of them. Following a button press the unchosen option disappeared immediately. After a 6 s delay approval (in the form of a thumbs up symbol) or disapproval (in the form of a thumbs down symbol) was revealed. To critically test for the role of positive and negative surprise, we also added 24 trials where feedback was not displayed (empty grey circle instead of a thumb symbol). After every 2–3 trials, participants were probed as to their current self-esteem (total of 78 self-esteem ratings) by being presented with the question 'How good do you feel about yourself at this moment?' for 5 s, after which they had 4 s to move a cursor along a visual analog scale with endpoints 'very bad' and 'very good'. Self-esteem probes were preceded by a jittered fixation display (uniformly distributed between 2 and 5 s). During instructions, we emphasized a distinction between self-esteem and mood (the former reflecting feeling good about yourself vs. the latter involving feeling good in general), self-esteem and self-concept (an affective evaluation of the self rather than 'cold' semantic knowledge about the self, e.g., 'I am a student', 'I am English'), and state vs. trait self-esteem (right now in the task vs. general feelings of self-worth reflecting the last weeks or months).

As part of a cover story participants were told that each rater made their ratings independently and that a visual color cue signaled how many profiles raters liked or disliked. For example, a rater who liked 36 out of 40 profiles would be placed in the first quartile group. To better orientate participants, they learned the rank ordering of the four groups prior to performing the task and were tested on the rank order before performing the task. In addition, explicit instructions emphasized that colors were unrelated to how a specific rater had evaluated them. To keep participants engaged they were told that for every trial where they failed to make a prediction before the time limit, 50 pence would be subtracted from a potential amount of money they would play with in a game after the scanning experiment (Dictator Game; see *Figure 2—figure supplement 2*). Missed trials were excluded from further analysis (the median number of missed trials was 1).

Unbeknownst to participants, social feedback was generated by the computer. All participants received positive feedback on 80 trials, negative feedback on 80 trials and no feedback on 24 trials. Importantly, the probability of receiving positive feedback was dependent on the rater's group. Participants received 'approval' feedback in 85%, 70%, 30%, and 15% of the trials, spanning the first to fourth quartile rater groups respectively. Participants were not informed about these exact probabilities before the task, but the percentages were consistent with information given to them about the order of the groups. Trial order was randomized such that participants never saw cues of the same color more than twice in a row, while the same color was never displayed more than seven trials in the past. The task was administered in three blocks with feedback that randomly ordered but always according with the 'approval frequency' of each group. Blocks lasted approximately 17 min in total. After each block participants received feedback about how many correct predictions they had made to increase engagement in the task.

## Additional experimental tasks

After scanning, participants performed two additional tasks: an 'Other evaluation' task (see *Figure 2—figure supplement 1*) and the Dictator Game (see *Figure 2—figure supplement 2*). The 'other evaluation' task was identical to the scanning task except that participants were not the object evaluation and that there were fewer trials (64 evaluations; by 16 raters of each group). In this task, participants were asked to predict whether another participant (of the same gender and age) was liked and then observed the feedback this other person received. After every 2–3 trials they reported on their own level of self-esteem (27 self-esteem ratings). For the Dictator Game, participants were endowed with £5. They played 12 independent Dictator games of which one was randomly selected for payout. They played a Dictator game with three members of each of the four

groups. To be specific, out of every group they played with: one person that approved of them, one person that disapproved of them and one person where no feedback was displayed. At the end of the experiment participants were told that the feedback was computer-generated.

## Computational models

We modeled dynamic changes in self-esteem for all ratings preceded by choices (74 in total) using exponential kernel regression models that assume an exponential decay of previous events. The winning model (*Equation 1*) contains separate terms for baseline self-esteem throughout and social approval prediction errors (*SPE*; the difference between received feedback and expected social approval from rater on each trial (*Equation 2*). Expectations about approval (ESV) were derived from participants' choice behavior in the task; see *Equation 3* below). The Gaussian noise term $\varepsilon \sim N(0, \sigma)$ allowed *Equation 1* to serve as a generative model of self-esteem. The influence of *SPE* was assumed to decay exponentially in time such that recent events had greater impact than earlier events (with forgetting factor: $0 < \gamma < 1$). *SPE*s on a given trial were operationalized as the difference between received social feedback and the *ESV*:

$$SPE^t = Social\ feedback - ESV^t \tag{2}$$

Where social feedback was 1 for approval, $-1$ for disapproval and 0 for 'no feedback'. ESV on each trial was estimated using a Rescorla-Wagner reinforcement-learning model (*Rescorla and Wagner, 1972*). This integrates information over trials by updating the *ESV* of raters from each of the four groups (k = 1–4) as follows:

$$ESV_k^{t+1} = ESV_k^t + \eta\ SPE^t \tag{3}$$

where $\eta$ is a learning rate capturing the weight that participants give to SPEs in updating expectations of social approval *ESV*. A softmax function transformed an *ESV* into an action probability for predicting to be liked or disliked:

$$\pi_L = \frac{1}{1 + e^{\frac{-(ESV+ESV_0)}{T}}} \tag{4}$$

Here $ESV_0$ is a response bias and $T$ a decision temperature parameter. A positive bias $ESV_0$ describes the 'extra credit' people give themselves, the willingness to predict being liked even in the absence of good evidence a rater will approve. Note that in our learning model the accumulation of $ESV$ itself is unbiased. Participants with larger $ESV_0$ persisted in predicting they would be liked by raters from groups for whom they had a negative $ESV$ (e.g., raters from groups 3 or 4). The decision temperature $T$ captures the 'motivational power of outcomes', i.e., the difference in $ESV$ that will increase the probability of predicting social approval by a fixed amount from the indifference point $\pi_L = 0.5$. Initial ESVs were two free parameters specifying initially expected approval rates for the most positive and the least positive group (i.e., $ESV_1^{(1)}$ and $ESV_4^{(1)}$).

Initial approval expectations for the other two groups were spaced equally in between:

$$ESV_2^{(1)} = ESV_1^{(1)} - \left(ESV_1^{(1)} - ESV_4^{(1)}\right)/3 \tag{5}$$

$$ESV_3^{(1)} = ESV_1^{(1)} - 2\left(ESV_1^{(1)} - ESV_4^{(1)}\right)/3 \tag{6}$$

For each individual participant all free parameters in *Equations 1-4* ($w_0$, $w_1$, $\sigma$, $\eta$, $ESV_0$, $T$, and initial ESVs) were fitted together so as to maximize the summed log-likelihood of self-esteem ratings and approval predictions. This model best explained choice behavior (i.e., predictions about approval) and changes in self-esteem in terms of how well the model described the data and its complexity (i.e., number of parameters) based on Bayesian model comparison. We compared this model against the following alternative models.

First, to justify the need of the response bias parameter, we compared model 1 to model 2 ('Learning, but no bias'), which is identical to model 1, but omits the response bias parameter $ESV_0$.

Second, to critically test whether a model that includes updating of expectations based on prediction errors weighted by a learning rate can better explain changes in self-esteem, we compared model 1 to model 3 ('Correct initial beliefs about approval'). This model assumes that participants do not update their expectations based on social feedback across the experiment, but start the experiment with the actual approval probabilities for each group. A comparison between model 3 and model 1 is critical given that participants were instructed about the rank order of the 4 groups prior to the experiment.

Third, to test whether expectations have an additional effect on changes in self-esteem above and beyond their effect captured by the SPE term, we compared the model against model 4 that had a separate expectations term based on existing models of prediction-error driven changes in subjective states (*Rutledge et al., 2014*; *2015*) (*Equation 7*).

$$\text{Self-esteem}(t) = w_0 + w_1 \sum_{j=1}^{t} \gamma^{t-j} \, ESV_j + w_2 \sum_{j=1}^{t} \gamma^{t-j} \, SPE_j + \epsilon \tag{7}$$

Finally, to verify that self-esteem not only depends on the valence of social feedback, but on errors arising out of expectations about feedback, we fitted model 6 to the self-esteem data ('Outcome valence only'; *Equation 8*).

$$\text{Self-esteem}(t) = w_0 + w_1 \sum_{j=1}^{t} \gamma^{t-j} SF_j + \epsilon \tag{8}$$

This model is most comparable to previous investigations of effects of social feedback on self-esteem where self-esteem is assumed to increase after positive social feedback (SF) and decrease after negative social feedback, but where expectations about approval are not modeled. Consistent with this notion, this model assumes that participants start the experiment without expectations about approval from raters from the four groups and do not update expectations based on feedback (i.e. expectations of .5 for each of the four groups).

To allow for a fair comparison between this model and our winning model, we fit this model to the self-esteem ratings only, because a model without expectations would by definition not provide a good fit for the behavioral choice data (as prediction choices clearly differed for the four groups). Therefore, we compared model 6 against models 5 that we also fitted to the self-esteem ratings only. Like model 6, model 5 assumed that participants did not update expectations. However, model 5 assumed that participants had initially expected approval rates for the most positive and the least positive group model that were specified by 2 free parameters and is therefore comparable to our winning model 1. A comparison of model 5 against this model 6 is critical for confirming that self-esteem is sensitive to prediction errors arising out of expectations about approval rather than merely to approval and disapproval per se.

## Model fitting and model comparison

To fit the parameters of the different computational models, we used maximum likelihood fitting with flat priors over the parameters. In order to examine whether models improved description of the experimental data we considered the mean squared error over self-esteem ratings. We considered the summed log likelihood of the model prediction over the predictions and the summed log density of the model prediction over the self-esteem ratings. To compare between models, we computed Bayesian Information Criterion (BIC) by penalizing the model evidence to account for model complexity as follows: $BIC = \ln(n)k - 2\ln(\hat{L})$, where $n$ is the number of choices + self esteem ratings used to compute the likelihood, $k$ is the number of fitted parameters and $\hat{L}$ is the maximized value of the likelihood function of the model. BIC measures are summed across all participants. Lower BIC values indicate a more parsimonious model fit.

## Canonical correlation analysis: Linking task parameters to symptoms

We performed CCA (*Hair et al., 1998*) to find how computational self-esteem parameters correlate with symptoms linked to low self-esteem. Symptoms were measured using questionnaires assessing self-evaluation, interpersonal sensitivity, as well as symptoms of psychiatric disorders characterized by negative self-beliefs. Self-evaluation measures included trait self-esteem (*Rosenberg, 1965*), state

self-esteem (*Heatherton and Polivy, 1991*), self-perception (*Neemann and Harter, 1986*), and narcissism (*Raskin and Terry, 1988*). Interpersonal sensitivity measures included the brief fear of negative evaluation scale (*Carleton et al., 2011*; *Leary, 1983*) and the rejection sensitivity questionnaire (*Downey and Feldman, 1996*). Symptom measures included state and trait anxiety (*Spielberger et al., 1970*), social anxiety (*Liebowitz, 1987*), and depression (*Angold et al., 1995*; *Beck et al., 1996*). Prior to running the CCA, aggregate questionnaire scores were z-scored across all participants and parameter estimates that were not normally distributed were log-transformed. For the CCA, we substituted the parameter estimates of initial expectations about the most positive and the least positive groups ($ESV_1^{(1)}$ and $ESV_4^{(1)}$) with two summary variables that carry the same information, but are more informative about the psychological processes involved (i.e., the average and the range of those two estimates).

## fMRI data acquisition

Scans were acquired using a 3T Siemens Trio MRI scanner (Siemens Healthcare, Erlangen, Germany) equipped with a standard transmit-receive 32-channel whole-head coil. After obtaining a localizer scan, we collected field maps (TE = 10 and 12.46 ms, TR = 102 ms, matrix size 64 × 64, with 64 slices, voxel size = 3 mm$^3$) for distortion correction. Subsequently, we acquired functional MRI data in three runs (mean amount of volumes = 1135; range 1105–1168; total number of volumes acquired varied depending on participants' choice times) with a blood oxygenation level-dependent (BOLD) sensitive T2*-weighted single shot echo-planar imaging (EPI) sequence (repetition time (TR) = 2.8 s, echo time (TE) = 30 ms, slice matrix = 64 × 64 × 40 matrix, slice thickness = 2 mm, slice gap = 1 mm gap, slice tilt of −30° (T > C), field of view (FOV) = 192 × 192 mm$^2$; ascending slice acquisition order), which was optimized to minimize signal dropout in ventral frontal and temporal cortex (*Deichmann et al., 2003*). The first five volumes from each functional run were discarded to allow for equilibration of T1 saturation effects. After the functional images, we obtained a 3D T1-weighted structural scan for anatomical reference (TR = 7.92 ms; TE = 2.48 ms, TI = 910 ms, flip angle α = 16°, 176 = slices, 1 × 1 × 1 mm voxels, FOV = 256 × 240 mm$^2$; *Deichmann et al., 2004*). We used a pulse-oximeter and breathing belt to collect physiological data during scanning.

Stimuli were presented in MATLAB (MathWorks, Inc., Natick, MA) using Cogent 2000 (Wellcome Trust Centre for Neuroimaging) onto a screen in the magnet bore, which participants could see through a mirror attached to the head coil. Participants could respond by using a fiber optic response box. During scanning foam inserts restricted head motion.

## fMRI analysis

Preprocessing and analysis of MRI data was implemented using Statistical Parametric Mapping 12 (SPM12) (Wellcome Trust Centre for Neuroimaging, University College London). Functional images were slice-time corrected, corrected using field maps, unwarped and realigned, co-registered with structural MRI, normalized to MNI space (using the DARTEL toolbox; *Ashburner, 2007*) and smoothed using a 8 mm, full-width at half-maximum isotropic Gaussian kernel.

To examine the neural correlates of SPEs and changes in self-esteem, we deployed two event-related general linear models (GLMs) that included separate regressors indicating onsets of choice, delay period, feedback, self-esteem question screen, and initial button press for self-esteem rating. All durations in the model were set to 0 s. The GLMs also contained 6 motion parameters regressors and 18 regressors for cardiac and respiratory regressors to correct for motion-induced and physiological noise.

The first GLM contained parametric modulators for 'ESV' (derived from computational modeling [*Equation 1*] and time-locked to choice onset), 'SPE' (derived from computational modeling and time-locked to feedback onset), and 'self-esteem rating' (z-scored and time-locked to question onset). The second GLM contained parametric modulators for 'inferred self-esteem rating' (derived from computational modeling; z-scored and time-locked to choice onset and onset of feedback), and actual 'self-esteem ratings' (z-scored and time-locked to question onset). Subject-specific contrast images were submitted to group level random-effects analyses. Results were corrected for multiple comparisons with Family-wise Error (FWE) cluster-correction at p<0.05 (cluster-forming threshold of p<0.005). To ensure that the FWE correction provided adequate control of false positives, we additionally performed non-parametric permutation tests (10,000 Monte-Carlo simulations)

that take into account the smoothness of the data and the normalized voxel size (*Slotnick et al., 2003*). These permutation tests determined that a cluster-extent threshold of k > 120 voxels is needed to control for multiple comparisons (p<0.05) at a cluster-forming threshold of p<0.005. All reported clusters exceed this threshold, supporting the validity of the FWE multiple-comparison correction procedure. We used the Marsbar toolbox (*Brett et al., 2002*; http://marsbar.sourceforce. net/) to extract subject-level contrast values in clusters of activity derived from our whole-brain analyses.

To examine functional connectivity, we performed a psychophysiological interaction (PPI) analysis (*Friston et al., 1997*; *O'Reilly et al., 2012*). We set up a GLM with regressors capturing the physiological effect, (i.e., time series for a 6 mm sphere centered on the peak voxel of the insula cluster [−44, 11, 9] derived from the whole-brain regression analysis testing for individual differences in SPE processing related to the interpersonal vulnerability dimension), the psychological contrast of interest (i.e., trial-by-trial self-esteem updates upon receipt of feedback) and the psychophysiological interaction term (i.e., physiological effect x psychological contrast of interest). The GLM also included 6 motion parameters regressors and 18 regressors for cardiac and respiratory regressors to correct for these sources of noise.

## Acknowledgements

The authors would like to thank Camilla Braine and Kalia Cleridou for their assistance during data collection and Eran Eldar and Tobias Hauser for discussions and comments on an earlier draft of the manuscript. GJW, MM and RJD are supported by the Neuroscience in Psychiatry Network, a strategic award by the Wellcome Trust to the University of Cambridge and University College London, of which RJD is a Principal Investigator (095844/Z/11/Z). RBR is supported by the Max Planck Society and a Medical Research Council Career Development Award (MR/N02401X/1). MM is supported by the Biomedical Research Council. RJD is supported by a Wellcome Trust Senior Investigator Award [098362/Z/12/Z]. The Wellcome Trust Centre for Neuroimaging is supported by core funding from the Wellcome Trust 091593/Z/10/Z]. The Max Planck UCL Centre for Computational Psychiatry and Ageing Research is funded jointly by the Max Planck Society and University College London.

## Additional information

### Funding

| Funder | Grant reference number | Author |
|---|---|---|
| Wellcome Trust | Strategic award (095844/Z/11/Z) | Geert-Jan Will<br>Michael Moutoussis<br>Raymond J Dolan |
| Medical Research Council | Career Development Award (MR/N02401X/1) | Robb B Rutledge |
| Wellcome Trust | Wellcome Trust Senior Investigator Award (098362/Z/12/Z) | Raymond J Dolan |
| Biomedical Research Council | | Michael Moutoussis |
| Max-Planck-Gesellschaft | UCL Centre for Computational Psychiatry and Ageing Research | Geert-Jan Will<br>Robb B Rutledge<br>Michael Moutoussis<br>Raymond J Dolan |
| Wellcome Trust | 091593/Z/10/Z | Geert-Jan Will<br>Robb B Rutledge<br>Michael Moutoussis<br>Raymond J Dolan |

The funders had no role in study design, data collection and interpretation, or the decision to submit the work for publication.

## Author contributions

Geert-Jan Will, Conception and design, Acquisition of data, Analysis and interpretation of data, Drafting or revising the article; Robb B Rutledge, Michael Moutoussis, Raymond J Dolan, Conception and design, Analysis and interpretation of data, Drafting or revising the article

## Author ORCIDs

Geert-Jan Will, http://orcid.org/0000-0003-1887-9829
Robb B Rutledge, https://orcid.org/0000-0001-7337-5039
Michael Moutoussis, https://orcid.org/0000-0002-4751-0425
Raymond J Dolan, http://orcid.org/0000-0001-9356-761X

## Ethics

Human subjects: Informed consent, and consent to publish, was obtained from every participant. Experimental procedures were approved by the Research Ethics Committee of University College London (UCL Ethics Project ID Number: 3450/002).

## Decision letter and Author response

Decision letter https://doi.org/10.7554/eLife.28098.014
Author response https://doi.org/10.7554/eLife.28098.015

# Additional files

## Supplementary files

• Supplementary file 1 Instructions and questions for online profile on which participants were evaluated
DOI: https://doi.org/10.7554/eLife.28098.011

• Supplementary file 2. Brain regions revealed by whole-brain neuroimaging analyses thresholded at p < .001 uncorrected, k > 50 voxels.
DOI: https://doi.org/10.7554/eLife.28098.012

• Transparent reporting form
DOI: https://doi.org/10.7554/eLife.28098.013

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
