## [Decision Letter]

Thank you for submitting your article "Neural and computational processes underlying dynamic changes in self-esteem" for consideration by *eLife*. Your article has been reviewed by two peer reviewers, who have opted to remain anonymous, and the evaluation has been overseen by Oriel FeldmanHall as Reviewing Editor and Richard Ivry as the Senior Editor. After extended discussion among the reviewers, Dr. FeldmanHall has drafted the following decision letter to provide a concise summary of the issues that must be addressed. As is reflected in his letter, it is not clear if a satisfactory response can be provided to all of the comments so you should give this careful thought in setting a course of action.

I was fortunate to get two very thoughtful reviews. Both reviewers think you are working on an important problem (as do I), and commented on how your study was both well designed, contained a good sample size, and that the manuscript was clearly written. However, both reviewers also had some important reservations with your interpretation of the data and the conceptual bases of your claims as well as various methodological aspects of the manuscript. I agree with their concerns, and while we all see potential in the manuscript, I am a bit on the fence on how to proceed since it is unclear whether the results of the study would change significantly due to the suggested changes, which could result in the manuscript ultimately being rejected.

While I have decided to encourage a response to these reviews, I leave it up to you to decide whether it is worth pursuing the recommendations of the reviewers. Perhaps the best way forward is to look at your data in light of the suggested changes, and then make a decision as to whether it makes sense to pursue this for publication in *eLife*.

Major issues.

1) A major concern raised by both reviewers was the way thresholding was carried out. The authors use an initial voxelwise cluster-forming threshold of p<0.005 uncorrected, before performing cluster correction. However, it has been demonstrated on many occasions that such a cluster-forming threshold is too liberal, potentially leading to false positives. The authors should set the cluster-forming threshold at p<0.001uncorrected and re-analyse the data. (Ekland et al., 2016, PNAS)

2) Both reviewers had trouble with the anatomical localizations on a number of fronts. It would be helpful if the authors could more clearly define the anatomical locations of their results and subsequently the relevant research that would support finding prediction error and update signals in these areas. For example, a previous study by Jones and colleagues required participants learn from social feedback of peers and found responses in rostral anterior cingulate cortex that tracked the modulation of expected values of receiving feedback from peers. They also found that ventral striatum signalled prediction errors (Jones et al., 2011).

2a) The peak coordinates reported for the 'ventral striatum' actually seem to be in either the subgenual anterior cingulate cortex (right peak) or ventral striatum/sgACC border (left peak) – it lies within area 25/the septal-hypothalamic area. Why does this matter? This region has been shown to signal specifically other-oriented prediction errors during social interactions (Diaconescou et al., 2016; Lockwood et al., 2016). Such an interpretation would of course be the opposite of that in the original manuscript – of domain-general mechanisms in the striatum – but would actually more closely align with the precise location of their result. In short, the anatomical interpretation of where these findings are should be revised in light of the anatomical location of their peak coordinates.

2b) The mPFC is also poorly characterised in the Introduction and Discussion, lacking in accuracy and specificity. This region contains several different zones that clearly do not all play the same roles in value and social information processing as implied (Neubert et al., 2015; Wittman et al., 2016; Apps et al., 2016; Nicolle et al., 2012). Moreover, the region identified in the results does not lie in the same portion of the VmPFC as that referred to in most of the papers cited. The peak lies within areas 24a/b or 32, which is superior and posterior to the typical "value" processing areas in the cited research. There is a significant body of research highlighting the roles of this particular region in coding social prediction errors and value signals during social interactions (Hill et al., 2016; Suzuki, 2012; Apps et al., 2016) and its links to social group size etc. (Sallet et al., 2009, Science), as well the more commonly discussed links to depressive symptoms. More specificity and engagement with the literature is required to make the discussion of their results connected with relevant extant literature.

3) Existing literature and scholarship. Relatedly, both reviewers pointed to inconsistencies with the existing literature. For example, the authors state “it is unclear how the brain does this with respect to evaluation of self” but several studies have investigated the tracking of self-evaluation, which has been consistently linked to vmPFC (e.g. Sui et al., 2013, PNAS; Kelley et al., 2002). Wittmann and colleagues have also found self related performance evaluation tracking in the perigenual anterior cingulate cortex, in a cluster that appears to overlap in part with the one reported here (Wittmann et al., 2016). Finally, in regards to 'Sociometer theory' – the discussion of this is rather unrelated to any of the findings, as none of the data directly tests the limits of this account. One suggestion is to remove this, as it didn't seem a particularly strong rebuttal of the account.

4) The last major issue that the reviewers highlighted was that it was unclear whether the update and SPE signals are orthogonal to the valence of the feedback. In studies of reinforcement learning the outcome and prediction error values can become highly correlated (Klein-Flugge et al., 2012), can the authors demonstrate this wasn't the case here? What is the correlation between simply an outcome valence parameter and the trial-by-trial values of SPE and update?

---

## [Author Response]

1) A major concern raised by both reviewers was the way thresholding was carried out. The authors use an initial voxelwise cluster-forming threshold of p<0.005 uncorrected, before performing cluster correction. However, it has been demonstrated on many occasions that such a cluster-forming threshold is too liberal, potentially leading to false positives. The authors should set the cluster-forming threshold at p<0.001uncorrected and re-analyse the data. (Ekland et al., 2016)

We agree with the reviewers that an increase in false positives due to low cluster-forming thresholds is a potentially serious issue. We have run additional analyses that confirm that this is not an issue for our study.

Eklund et al., 2016 argue that the most reliable way to control for false positives are non-parametric permutation tests. In line with this recommendation, we ran a bias-free data-driven permutation test (Slotnick et al., 2003). This test indicated that to control the family-wise error rate (p <.05) at a cluster-forming threshold of p < 0.005, with our specific acquisition and smoothing parameters, a cluster needs to comprise at least 120 voxels. All clusters we report in our paper exceed this threshold. Thus, there are no principled concerns in relation to the threshold and the inclusion of these permutation tests strengthens our results.

The referees will be reassured that all reported results remain significant with a cluster-forming threshold of p <.001, except for the neural representation of trial-by-trial self-esteem updates in vmPFC. However, given that the vmPFC cluster we report (868 voxels) is 7-fold larger than the cluster extent required based on the non-biased permutation tests, this gives us a high degree of confidence that this finding is not a false positive.

2) Both reviewers had trouble with the anatomical localizations on a number of fronts. It would be helpful if the authors could more clearly define the anatomical locations of their results and subsequently the relevant research that would support finding prediction error and update signals in these areas. For example, a previous study by Jones and colleagues required participants learn from social feedback of peers and found responses in rostral anterior cingulate cortex that tracked the modulation of expected values of receiving feedback from peers. They also found that ventral striatum signalled prediction errors (Jones et al., 2011).2a) The peak coordinates reported for the 'ventral striatum' actually seem to be in either the subgenual anterior cingulate cortex (right peak) or ventral striatum/sgACC border (left peak) – it lies within area 25/ the septal-hypothalamic area. Why does this matter? This region has been shown to signal specifically other-oriented prediction errors during social interactions (Diaconescou et al., 2016; Lockwood et al., 2016). Such an interpretation would of course be the opposite of that in the original manuscript – of domain-general mechanisms in the striatum – but would actually more closely align with the precise location of their result. In short, the anatomical interpretation of where these findings are should be revised in light of the anatomical location of their peak coordinates.

We thank the reviewers for these valuable suggestions, which we trust has now led to an improvement to our Introduction and Discussion. The reviewers are right that our cluster in bilateral ventral striatum includes not only the ventral striatum but also extends into subgenual ACC, an area that is correlated with other-oriented prediction errors. We have revised our Discussion in light of interesting findings of Diaconescou et al., 2017; Lockwood et al., 2016 and Jones et al., 2011, 2014) as follows:

“The striatum encodes prediction errors in learning about primary (D'Ardenne et al., 2008; Hart et al., 2014) and secondary rewards (Caplin et al., 2010; Pessiglione et al,. 2006), including learning about social acceptance (Jones et al., 2011; Jones et al., 2014). […] The presence of a cluster spanning both striatum and sgACC in response to this multiplexed prediction error signal dovetails with these prior findings.”

We agree that the extent of activation in relation to these prior studies does not allow us to exclude an exclusively social mechanism. Therefore, we have now removed the interpretation of our findings as evidence for “domain general” mechanisms. It will be a goal for future research to disentangle the different contributions of ventral striatum and sgACC during learning about the self during interactions with others.

2b) The mPFC is also poorly characterised in the Introduction and Discussion, lacking in accuracy and specificity. This region contains several different zones that clearly do not all play the same roles in value and social information processing as implied (Neubert et al., 2015; Wittman et al., 2016; Apps et al., 2016; Nicolle et al., 2012). Moreover, the region identified in the results does not lie in the same portion of the VmPFC as that referred to in most of the papers cited. The peak lies within areas 24a/b or 32, which is superior and posterior to the typical "value" processing areas in the cited research. There is a significant body of research highlighting the roles of this particular region in coding social prediction errors and value signals during social interactions (Hill et al., 2016; Suzuki, 2012; Apps et al., 2016) and its links to social group size etc. (Sallet et al., 2009, Science), as well the more commonly discussed links to depressive symptoms. More specificity and engagement with the literature is required to make the discussion of their results connected with relevant extant literature.

We thank the reviewer for emphasizing a need to acknowledge this relevant literature, which we now include in our Introduction and Discussion. We revised the Introduction such that it addresses the different subregions of mPFC and their involvement in valuation and social learning specifically as follows:

“A candidate neural substrate for integrating social evaluation with self-evaluation is the ventromedial Prefrontal Cortex (vmPFC) given evidence that being evaluated by others (Dalgleish et al., 2017; Gunther Moor et al., 2010; Somerville, Kelley and Heatherton, 2010), and evaluating the self (Chavez, Heatherton and Wagner, 2016; D'Argembeau et al., 2011; Hughes and Beer, 2013; Kelley et al., 2002) activates subregions of vmPFC. The vmPFC comprises multiple distinct cytoarchitectonic zones including parts of the anterior cingulate cortex (ACC; including Brodmann areas 24 and 32) and orbitofrontal cortex (OFC; including Brodmann areas 11, 13, 14) (Haber and Knutson, 2010). These subregions have distinct connectivity profiles (Neubert et al., 2015) and are thought to show specialization for cognitive functions important in valuation (Bartra, McGuire and Kable, 2013; FitzGerald, Seymour and Dolan, 2009; Lebreton et al., 2015), social learning (Apps, Rushworth and Chang, 2016; Hill, Boorman and Fried, 2016; Suzuki et al., 2012) and their intersection (Apps and Ramnani, 2017; Garvert et al., 2015; Nicolle et al., 2012).”

We used the “Cingulate and orbitofrontal cortex” atlas created by Neubert et al., 2015 to label our peaks in medial PFC in the Results section. Based on this more precise characterization of subregions in mPFC we have revised our Discussion as follows:

“Our results show that updates of self-worth are represented in a cluster in vmPFC with a peak in medial OFC (BA 14m), extending into pgACC (BA 32pl). […] Adjacent subregions in vmPFC may thus integrate separate strands of information about current value assigned to the self in order to estimate self-value in the future.”

3) Existing literature and scholarship. Relatedly, both reviewers pointed to inconsistencies with the existing literature. For example, the authors state “it is unclear how the brain does this with respect to evaluation of self” but several studies have investigated the tracking of self evaluation, which has been consistently linked to vmPFC (e.g. Sui et al., 2013, PNAS; Kelley et al., 2002). Wittmann and colleagues have also found self related performance evaluation tracking in the perigenual anterior cingulate cortex, in a cluster that appears to overlap in part with the one reported here (Wittmann et al., 2016). Finally, in regards to 'Sociometer theory' – the discussion of this is rather unrelated to any of the findings, as none of the data directly tests the limits of this account. One suggestion is to remove this, as it didn't seem a particularly strong rebuttal of the account.

We agree with the reviewers that the claim that “It is unclear how the brain does this with respect to the evaluation of the self” can be misinterpreted.

We have now removed this sentence from our Introduction.

To the best of our knowledge our study is the first to characterize the computational and neural processes underlying experimentally induced changes in self-esteem in response to social evaluation. It builds on prior work on the neural correlates of self-evaluation such as that of Kelly et al., 2002;, Sui et al., 2013 and Wittman et al., 2016. Kelley et al., 2002 examined neural correlates of self-evaluation vs. other-evaluation using a task where participants had to judge whether an adjective described them or another person. Sui et al., 2013 investigated the neural correlates of learning to associate a neutral stimulus with either the self or other people. These studies were pivotal in identifying the medial PFC as the substrate for self-evaluations, but do not formally address the question of how self-evaluative beliefs are updated and how this is implemented in the brain.

Wittman et al. showed that perigenual ACC tracks the history of successes in a social context (Wittmann et al., 2016), where activity in this region was higher in individuals whose subjective evaluation of their performance in the task was affected more strongly by their performance. We build on these prior findings by showing this brain region is involved in updating a more general value ascribed to the self upon learning how we are valued by others. Additionally, we show that connectivity between this region and the insula, correlates with interpersonal vulnerability, providing a potential biomarker for psychiatric vulnerability.

We now clarify these issues:

“A recent study showed that the perigenual ACC tracks a history of one’s own success and failures in a social context, while dorsomedial prefrontal area 9 tracked performance history of an interaction partner (Wittmann et al., 2016). Crucially, activity in pgACC was higher in individuals whose subjective evaluation of their own performance was affected more strongly by their actual performance, making this region a candidate substrate for online updating of self-esteem.”

We apologize for the ambiguous language with respect to sociometer theory.

Our results confirm predictions made by sociometer theory and provide mechanistic insight at both the algorithmic and neural level. Our results further lead to an important new hypothesis following on from sociometer theory, namely that self-esteem is a learning signal used to learn about our social standing in new environments. We have clarified this in our Discussion as follows:

“Our study confirms predictions made by sociometer theory. […]These findings lead to a new testable hypothesis, namely that self-esteem encompasses a form of learning signal that we use to gauge our social standing in new environments.”

4) The last major issue that the reviewers highlighted was that it was unclear whether the update and SPE signals are orthogonal to the valence of the feedback. In studies of reinforcement learning the outcome and prediction error values can become highly correlated (Klein-Flugge et al., 2012), can the authors demonstrate this wasn't the case here? What is the correlation between simply an outcome valence parameter and the trial-by-trial values of SPE and update?

This is an important issue that the reviewers highlight. The outcome and prediction error are, by definition, highly correlated because the prediction error is the difference between the (valence of the) outcome and the expectation. This is common to all reinforcement learning experiments.

Fortunately, there are several ways to demonstrate that the SPE model provides a better explanation for our results than a model based on outcome valence alone. First, we compared our prediction error model against an “Outcome valence only” model (Model 6; see Table 1). Our prediction error model explains 13% more of the variance in self-esteem ratings (median r^2^=0.31) than is the case for a “valence model” (median r^2^=0.18). This was also confirmed by Bayesian model comparison (90 BIC point difference).

Second, if self-esteem only varied in response to the valence of feedback, no relationship would exist with expectations after regressing out outcome valence. A prediction error is defined as the outcome minus the expectation. Therefore, if social approval prediction errors explain changes in self-esteem better than outcome valence alone, trial-by-trial changes in self-esteem should correlate positively with outcome valence and negatively with expectations (Behrens, et al., 2008). Regression analyses showed that this was indeed the case. We first regressed outcome valence for every trial against subsequent self-esteem ratings (z-scored). This resulted in a positive correlation on average, r=0.18 across subjects (p < 1 × 10^-9^). Subsequently, we regressed z-scored expectations against the residuals of the previous regression model. This analysis showed that even after accounting for the variance explained by outcome valence, there was a significant negative correlation on average with expectations, r= -0.06 (p =.012).

These analyses provide quantitative evidence that dynamic changes in self-esteem are better explained by social approval prediction errors than merely the valence of social feedback. We have included the additional analysis:

“Next, we tested the hypothesis that dynamic changes in self-esteem depend on both the valence of social feedback and expectations about feedback. If social approval prediction errors explain changes in self-esteem better than outcome valence alone, trial-by-trial changes in self-esteem should correlate positively with outcome valence and negatively with expectations (Behrens et al., 2008). We found that, after regressing out the effects of outcome valence (r=0.18, p < 1 × 10^-9^), there remained a significant negative correlation with expectations (r= -0.06; p=0.012) (see Figure 2).

We have clarified that we used Bayesian model comparison to compare our model to an “outcome valence only” model as follows:

“We chose this model as it outperformed a range of alternative plausible models, including a model that took the valence of feedback into account, but did not feature expectations about approval (“Outcome valence only” model 6; see Table 1).”